

# Developmental effects of environmental light on male nuptial coloration in Lake Victoria cichlid fish

Daniel Shane Wright[1], Emma Rietveld[1,2] and Martine E. Maan[1]

[1] Groningen Institute for Evolutionary Life Sciences, University of Groningen, Groningen, Netherlands
[2] University of Applied Sciences van Hall Larenstein, Leeuwarden, Netherlands

Corresponding author
Daniel Shane Wright,
d.s.wright@rug.nl

## ABSTRACT

**Background**. Efficient communication requires that signals are well transmitted and perceived in a given environment. Natural selection therefore drives the evolution of different signals in different environments. In addition, environmental heterogeneity at small spatial or temporal scales may favour phenotypic plasticity in signaling traits, as plasticity may allow rapid adjustment of signal expression to optimize transmission. In this study, we explore signal plasticity in the nuptial coloration of Lake Victoria cichlids, *Pundamilia pundamilia* and *Pundamilia nyererei*. These two species differ in male coloration, which mediates species-assortative mating. They occur in adjacent depth ranges with different light environments. Given the close proximity of their habitats, overlapping at some locations, plasticity in male coloration could contribute to male reproductive success but interfere with reproductive isolation.

**Methods**. We reared *P. pundamilia*, *P. nyererei*, and their hybrids under light conditions mimicking the two depth ranges in Lake Victoria. From photographs, we quantified the nuptial coloration of males, spanning the entire visible spectrum. In experiment 1, we examined developmental colour plasticity by comparing sibling males reared in each light condition. In experiment 2, we assessed colour plasticity in adulthood, by switching adult males between conditions and tracking coloration for 100 days.

**Results**. We found that nuptial colour in *Pundamilia* did respond plastically to our light manipulations, but only in a limited hue range. Fish that were reared in light conditions mimicking the deeper habitat were significantly greener than those in conditions mimicking shallow waters. The species-specific nuptial colours (blue and red) did not change. When moved to the opposing light condition as adults, males did not change colour.

**Discussion**. Our results show that species-specific nuptial colours, which are subject to strong divergent selection by female choice, are not plastic. We do find plasticity in green coloration, a response that may contribute to visual conspicuousness in darker, red-shifted light environments. These results suggest that light-environment-induced plasticity in male nuptial coloration in *P. pundamilia* and *P. nyererei* is limited and does not interfere with reproductive isolation.

## INTRODUCTION

Natural selection favors communication signals that maximize reception and minimize degradation (*Endler, 1992*). Environmental heterogeneity can alter signal transmission, resulting in signal variation across environments (*Endler, 1990*; *Endler, 1992*). The link between colour signals and local light conditions is well established (as reviewed by: *Endler & Mappes, 2017*), with many examples particularly from aquatic organisms (*Seehausen, Van Alphen & Witte, 1997*; *Boughman, 2001*; *Fuller, 2002*; *Cummings, 2007*; *Morrongiello et al., 2010*; *Kelley et al., 2012*). However, changing environmental conditions could disrupt these relationships, rendering previously conspicuous signals ineffective. In such instances, flexibility in colour signaling may prove beneficial and recent work has documented this capacity in a number of fish species (killifish: *Fuller & Travis, 2004*; sticklebacks: *Lewandowski & Boughman, 2008*; tilapia: *Hornsby et al., 2013*).

Plasticity in mating signals can have major evolutionary consequences. In particular, when signals mediate reproductive isolation, plastic changes in response to environmental variation could affect the extent of assortative mating, resulting in gene flow that may inhibit or even reverse species differentiation. Conversely, plasticity in mating signals can also provide a starting point for species divergence, as has been suggested for song learning in birds (*Lachlan & Servedio, 2004*; *Mason et al., 2017*). Here, we examine how changes in the local light environment affect colour signaling in Lake Victoria cichlids.

In teleost fish, coloration derives from cells specialized for the storage and synthesis of light-absorbing and light-reflecting structures (*Sugimoto, 2002*; *Leclercq, Taylor & Migaud, 2010*). These cells, chromatophores, are distributed throughout the integument and are responsible for the wide variety of colours and patterns present in fish (*Leclercq, Taylor & Migaud, 2010*). In addition to genetic variation, fish coloration may change plastically in response to a multitude of factors (e.g., nutritional state, social interactions, local conditions, *Leclercq, Taylor & Migaud, 2010*). Short-term (physiological) colour change— e.g., in signaling social state (*Maan & Sefc, 2013*)—involves hormonal and neurological processes that affect the density of pigments within existing chromatophores (*Sugimoto, 2002*). Over longer time scales (e.g., across development), fish can undergo colour change by the generation of new and/or the death of existing chromatophores (*Sugimoto, 2002*). Both processes are likely to play a role in the adjustment of colour signals to changing environmental conditions.

*Pundamilia pundamilia* (*Seehausen et al., 1998*) and *Pundamilia nyererei* (*Witte-Maas & Witte, 1985*) are two closely related, rock-dwelling species of cichlid fish that co-occur at rocky islands in southern Lake Victoria (*Seehausen, 1996*). They are anatomically very similar and behave as biological species in clear waters but hybridize in more turbid waters (*Seehausen, Van Alphen & Witte, 1997*). Males of the two species are distinguished by their nuptial coloration; *P. pundamilia* males are blue/grey, whereas *P. nyererei* males are yellow with a crimson-red dorsum. Females of both species are yellow/grey in colour (*Seehausen, 1996*; *Van Alphen, Seehausen & Galis, 2004*). Although sympatric, the two species tend to have different depth distributions: *P. pundamila* is found in shallower waters while *P. nyererei* extends to greater depths. High turbidity in Lake Victoria results

in a shift of the light spectrum toward longer wavelengths with increasing depth and, as such, *P. nyererei* inhabits an environment largely devoid of short-wavelength light (*Maan et al., 2006*; *Seehausen et al., 2008*; *Castillo Cajas et al., 2012*). Previous work has found female preferences for conspecific male nuptial colouration in both species (*Seehausen & Van Alphen, 1998*; *Haesler & Seehausen, 2005*; *Stelkens et al., 2008*; *Selz et al., 2014*) and the differences in male colour are necessary and sufficient for reproductive isolation (*Selz et al., 2014*). However, we have recently observed that female preferences are influenced by the light environment experienced during development (*Wright et al., 2017*). When reared in broad-spectrum light, characteristic of the *P. pundamila* habitat, females more often preferred the blue *P. pundamilia* males while females reared in red-shifted light, characteristic of *P. nyererei* habitats, tended to prefer the red *P. nyererei* males (*Wright et al., 2017*). Given the role of the light environment in female preference determination, a question then follows: *how does the local light environment affect the expression of male nuptial colour*?

Observations from wild populations suggest that the local light environment does influence coloration, as *P. nyererei* from turbid (long wavelength-shifted) and clear water (broad-spectrum) locations differ in redness (*Maan, Seehausen & Van Alphen, 2010*; *Castillo Cajas et al., 2012*). Anal fin spots, characteristic yellow-orange ovoid markings on the anal fins of Haplochromine cichlids (*Goldschmidt, 1991*; *Maan & Sefc, 2013*), also co-vary with environmental light. *Goldschmidt (1991)* reported that Lake Victoria species inhabiting darker environments have larger anal fin spots and, more recently, Theis and colleagues reported that *A. burtoni* from Lake Tanganyika have less intensely coloured spots than populations from turbid rivers (*Theis et al., 2017*). These patterns are implicitly attributed to genetic variation, but phenotypic plasticity may also play a role. With the close proximity of *P. pundamilia* and *P. nyererei* habitats (a few meters to tens of meters, with overlapping distributions at several locations: *Seehausen et al., 2008*) and the fact that light conditions can fluctuate between seasons and due to weather (wind/rain), selection may favour some degree of plasticity in male colour expression. In fact, plasticity in cichlid colour has been documented: Nile tilapia increased short-wavelength body reflectance when reared under red-shifted light (*Hornsby et al., 2013*) and both South American (*Kop & Durmaz, 2008*) and African cichlids (*McNeil et al., 2016*) changed colour in response to carotenoid availability in the diet. Diet-induced colour changes have also been observed in *Pundamilia* (ME Maan, pers. obs., 2012; DS Wright, pers. obs., 2015), but common-garden and breeding experiments suggest strong heritability and low plasticity of the interspecific colour differences (*Magalhaes et al., 2009*; *Magalhaes & Seehausen, 2010*).

In this study, we experimentally manipulated environmental light and tested its effect on male nuptial colour expression. By rearing sibling males under light conditions mimicking shallow and deep habitats of Lake Victoria, we were able to ask: *does the light environment experienced during ontogeny affect the development of nuptial coloration in Pundamilia*? Given that blue colour is an ineffective signal in deep-water light conditions (lacking short wavelengths), we predicted that deep-reared fish might exhibit more long-wavelength reflecting coloration. We also moved a sub-set of males between rearing environments during adulthood, allowing us to test the effect of sudden environmental change and ask:

*do adult Pundamilia males adjust their colour in response to changing conditions*? Again, we predicted that fish moved to deep light would express more long-wavelength reflecting colours.

## METHODS

### Fish rearing & maintenance

Offspring of wild caught *P. pundamilia* and *P. nyererei*, collected at Python Islands in the Mwanza Gulf of Lake Victoria (−2.6237, 32.8567 in 2010 & 2014), were reared in light conditions mimicking those in shallow and deep waters at Python Islands (as in: *Maan et al., 2017*; *Wright et al., 2017*). Lab-bred lines (hybrid and non-hybrid) were created opportunistically as reciprocal crosses, with 18 dams and 14 sires. Hybridization does occur with low frequency at Python Islands (*Seehausen et al., 2008*) and can be accomplished in the lab by housing females with heterospecific males. Fourteen F1 crosses (wild parents: 6 *P. nye x P. nye;* 4 *P. pun x P. pun*; 1 *P. nye x P. pun*; 3 *P. pun x P. nye)* and five F2 crosses (lab-bred parents: 1 *P. nye x P. pun*; 4 *hybrid* x *hybrid*) resulted in a test population of 58 males from 19 families (family details provided in Table S1). We included F2 fish due to low availability of F1 hybrids.

*Pundamilia* are maternal mouth brooders; fertilized eggs were removed from brooding females approximately six days after spawning (mean ± se: 6.3 ± 0.5 days post-fertilization; eggs hatch at about 5–6 dpf) and split evenly between light conditions. Upon reaching maturity, males displaying nuptial coloration were removed from family groups, PIT tagged (Passive Integrated Transponders, from Biomark, Idaho, USA, and Dorset Identification, Aalten, The Netherlands), and housed individually, separated by transparent, plastic dividers. All males were housed next to a randomly assigned male, with either one or two neighbour males (depending on location within the tank). Neighboring fish were the same for the duration of each sampling period (more details below). Fish were maintained at 25 ± 1 °C on a 12L: 12D light cycle and fed daily a mixture of commercial cichlid flakes and pellets and frozen food (artemia, krill, spirulina, black and red mosquito larvae). This study was conducted under the approval of the Institutional Animal Care and Use Committee of the University of Groningen (DEC 6205B; CCD 105002016464). The Tanzania Commission for Science and Technology (COSTECH) approved field permits for the collection of wild fish (2010-100-NA-2010-53 & 2013-253-NA-2014-177).

### Experimental light conditions

Experimental light conditions were created to mimic the natural light environments of *P. pundamilia* and *P. nyererei* at Python Islands, Lake Victoria (described in greater detail: *Maan et al., 2017*; *Wright et al., 2017*). Species-specific light spectra were simulated in the laboratory (Fig. S1) by halogen light bulbs filtered with a green light filter (LEE #243; Lee Filters, Andover, UK). In the 'shallow' condition, mimicking *P. pundamilia* habitat, the spectrum was blue- supplemented with *Paulmann* 88090 compact fluorescent 15W bulbs. In the 'deep condition', mimicking *P. nyererei* habitat, short wavelength light was reduced by adding a yellow light filter (LEE #015). The light intensity differences between depth ranges in Lake Victoria are variable and can change rapidly depending on weather and sun angle

(as much as 1,000-fold in sun vs. cloud cover); the mean ($\pm$se) light intensity in the deep environment (measured in 2010) was $34.15 \pm 3.59\%$ of that in the shallow environment (Fig. S1). Our experimental light conditions were designed to mimic in particular the spectral differences between depths and only partly recreated the intensity difference (the deep condition had a light intensity of $\sim$70% of that of the shallow condition).

### Experiment 1: developmental colour plasticity

Males reared under experimental light conditions from birth were photographed repeatedly (three times each) in adulthood and assessed for body/fin coloration (details below). In total, we examined 29 pairs of brothers (mean age $\pm$ se at first sample: $689.9 \pm 67$ days; *Pundamilia* reach sexual maturity at $\sim$240 days), 29 from each light condition ($2 \times 10$ *P. pun*, $2 \times 9$ *P. nye*, $2 \times 10$ hybrids, Table S1). Males were sampled from August–October 2016, with a mean ($\pm$se) of $13.25 \pm 0.83$ days between samples. Neighbour males (those housed next to test fish) were maintained for the duration of the sampling period.

### Experiment 2: colour plasticity in adulthood

Following experiment 1, a subset of fish (Table S2) was switched to the opposing light condition (mean age $\pm$ se when switched: $643.47 \pm 50.61$ days; sexual maturity is $\sim$240 days) and colour tracked for 100 days. Each fish was photographed 11 times over the 100-day period: 1, 2, 3, 4, 7, 10, 14, 18, 46, 73, 100 days after switching. We switched 24 males, 12 from each light condition ($2 \times 4$ *P. pun*, $2 \times 4$ *P. nye*, $2 \times 4$ hybrid). As a control, we also tracked 18 males (nine from each light condition: $2 \times 3$ *P. pun*, $2 \times 3$ *P. nye*, $2 \times 3$ hybrid) that remained in their original rearing light, but were moved to different aquaria (thus, both experimental and control fish had new 'neighbour' males). All fish, control and experimental, were photographed at the same 11 time points (in addition to the three photographs from experiment 1). The experiment was conducted in two rounds: October 2016–January 2017 (24 fish moved: six experimental & six control from each light condition) and December 2016–March 2017 (18 fish moved: six experimental & three control from each light condition).

### Photography

All males were photographed under standardized conditions with a Nikon D5000 camera and a Nikon AF-S NIKKOR18-200 mm ED VR II lens. Fish were removed from their housing tank and transferred to a glass cuvette, placed within a 62.5 cm $\times$ 62.5 cm domed photography tent (Kaiser Light Tent Dome-Studio). This tent ensured equal illumination for all photos provided by an external flash (Nikon Speedlight SB-600) set outside of the tent. To ensure consistency of colour extracted from digital images (*Stevens et al., 2007*), all photos contained a grey and white standard attached to the front of the cuvette (Kodak colour separation guide), were taken with the same settings (ISO: 200; aperture: F9; exposure: 1/200; flash intensity: 1/8), and saved in RAW format.

### Colour analysis

In Adobe Photoshop CS4, we adjusted the white balance and removed the background from each photo, keeping the entire fish (except the eye and pelvic fins). Each fish was then

cropped into separate sections (body excluding fins, dorsal fin, caudal fin, anal fin, anal fin spots) and saved as individual images. Each section was analyzed for coloration using ImageJ (https://imagej.nih.gov/ij/), following the same procedure as detailed in *Selz et al. (2016)*. We defined specific colours by their individual components of hue, saturation, and brightness to cover the entire hue range, resulting in a measure of the number of pixels that met the criteria for red, orange, yellow, green, blue, magenta, violet, and black for each section (colour parameter details provided in Table S3).

### Brightness

We also measured the mean brightness of fish. Using Photoshop, we recorded the *luminosity* of 'whole fish' and 'anal fin spot' images, calculated from RGB values as: $0.3R + 0.59G + 0.11B$ (defined as *brightness* in: *Bockstein, 1986*). The weighting factors used by Photoshop (0.3, 0.59, 0.11) are based on human perception and should be similar to the trichromatic visual system of *Pundamilia* (*Carleton et al., 2005*). We measured the mean brightness of all fish used in experiment 1 and from three time points in experiment 2 (days 1, 10, 100).

## STATISTICAL ANALYSIS

### Colour scores

Colour scores were defined as a percentage of coverage: the number of pixels in each colour category divided by the total number of pixels in the section. We used principal component analysis (PCA) on the correlation matrix of all eight colour scores to obtain composite variables of coloration (separate PCA was performed for each section—loading matrices in Table S4). In experiment 1, we examined PC1–PC4, as PC5 accounted for <10% of the variance in all analyses (mean cumulative variance = 82.5%; mean across all sections). For all analyses, we first assessed 'whole fish' images (minus eye and pelvic fins), followed by examination of each individual section (body, dorsal fin, caudal fin, anal fin, anal fin spots). Anal fin spots contained only red, orange, and yellow, thus PC's were based on only those colour scores (and consequently, only PC1 & PC2 were used in analyses, 96.8% cumulative variance, Table S4).

In experiment 2, we first calculated baseline mean PC scores per fish using the repeated samples from experiment 1. At each time point after the switch, we then assessed deviation from the mean, calculated as: PC score—mean baseline PC score. Measuring the deviations from individual means allowed us to track the direction of colour change for each fish, independent of individual variation in baseline. Once again, PC scores were calculated for each body part independently and we used only PC1–PC4 (mean cumulative variance = 79.8%; loading matrices in Table S5).

### Experiment 1: developmental colour plasticity

Using linear mixed modeling (lmer function in the lme4 package, *Bates et al., 2013*) in R (v3.3.2; *R Development Core Team, 2016*), we tested PC's for the influence (and interactions) of: rearing light (shallow vs. deep), species (*P. pun*, *P. nye*, or hybrid), and body size (standard length, SL). Random effects included fish identity, parental identity, aquaria number, and position within aquaria to account for: (1) repeated sampling,

(2) shared parentage among fish (Table S1), (3) location of aquaria within the housing facility, (4) number of neighboring males (1 or 2). The optimal random effect structure of models was determined by AIC comparison (*Sakamoto, Ishiguro & Kitagawa, 1986*) and the significance of fixed effect parameters was determined by likelihood ratio tests (LRT) via the *drop1* function. Minimum adequate statistical models (MAM) were selected using statistical significance (*Crawley, 2002*; *Nakagawa & Cuthill, 2007*). We then used the *KRmodcomp—pbkrtest* package (*Halekoh & Højsgaard, 2014*) to test the MAM against a model lacking the significant parameter(s), which allowed us to obtain the estimated effect size of fixed effect parameters under the Kenward–Roger (KR) approximation (*Kenward & Roger, 1997*; *Kenward & Roger, 2009*). In the case of more than two categories per fixed effect parameter (i.e., species), we used post hoc Tukey (glht—multcomp package: *Hothorn, Bretz & Westfall, 2008*) to obtain parameter estimates.

### Anal fin spot number

Following *Albertson et al. (2014)*, the number of anal fin spots was counted as the sum of complete (1.0 each) and incomplete (0.5 each) spots for each fish (incomplete fin spots occur along the perimeter of the anal fin, often becoming complete with age/growth). Total spot number was compared among species, rearing light, and SL using the *glmer.nb* function in *lme4* (*Bates et al., 2013*). Random effects were the same as above and reduction to MAM followed the same procedure. As *KRmodcomp* is unavailable for *glmer.nb*, final parameter estimates are reported from LRT via the *drop1* function (*Ripley et al., 2015*).

### Experiment 2: colour plasticity in adulthood

Using *lme* in package *nlme* (*Pinheiro et al., 2014*), we tracked fish coloration change over time, testing the influence (and interactions) of: species, treatment (rearing environment + 'switched' environment) and date (of sampling). We used *lme* because it allows specification of the optimal autocorrelation structure, as autocorrelation is common in longitudinal data (*Crawley, 2007*; *Zuur et al., 2009*). Random effects were the same as above, but with an additional random slope/random intercept term for date and fish identity (∼date|fish identity) to account for variability in the nature of colour change over time between individual fish. For simplification to MAM, models were fit with maximum likelihood (ML) and selected for statistical significance (*Crawley, 2002*; *Nakagawa & Cuthill, 2007*) by LRT using *drop1*. Final models were refit with restricted maximum likelihood (REML) and fixed effect parameters of MAM reported from the *anova* function. As above, we used post hoc Tukey (*Hothorn, Bretz & Westfall, 2008*) to obtain estimates for more than two categories per parameter.

## RESULTS

### Light-independent, interspecific differences
#### Coloration

To estimate the overall 'colourfulness' of fish, we calculated the sum of all measured colour scores for each male (whole body). Species did not differ in colourfulness ($P = 0.29$), nor did they differ in colours *not* defined by our colour parameters (calculated as: 100—sum of all measured colours; $P = 0.29$).

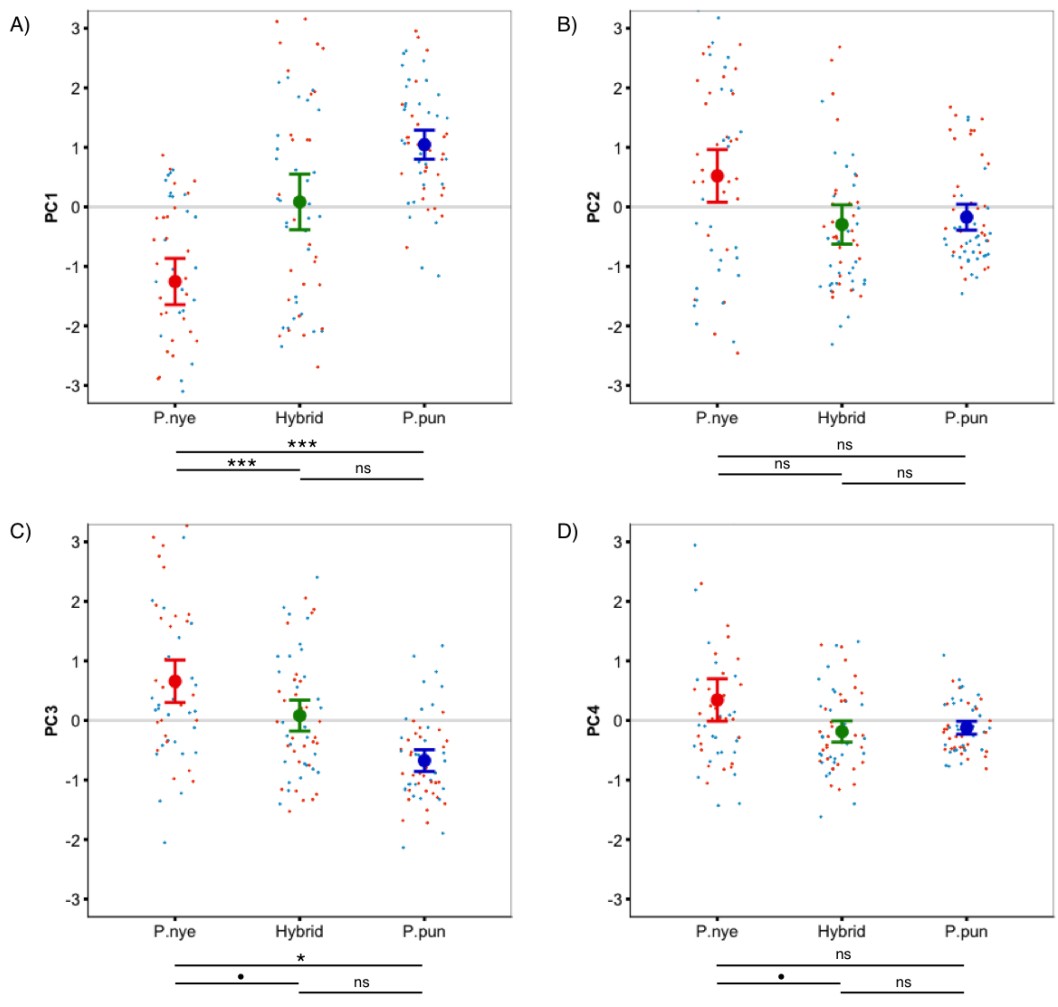

**Figure 1 Species colour differences.** Species-specific scores for 'whole fish' coloration, expressed as principal components. Linear mixed modeling revealed significant species differences for PC1 (A), PC3 (C), and PC4 (D), but not for PC2 (B). Points represent individual PC scores, coloured as deep (orange) and shallow (light blue) rearing light. Error bars represent 95% CI; ● indicates $P < 0.1$, * indicates $P < 0.05$, ** indicates $P < 0.01$, *** indicates $P < 0.001$.

There was a significant difference between species ($F_{2,55.00} = 13.40$, $P < 0.001$, Fig. 1A) in whole fish PC1 (positive loading yellow/orange). Tukey post hoc revealed that *P. nyererei* scored significantly lower than *P. pundamilia* ($Z = -5.39$, $P < 0.001$) and hybrids ($Z = -3.76$, $P < 0.001$). *P. pundamilia* was highest but did not differ significantly from hybrids ($P = 0.47$). There were tendencies for differences among species for whole fish PC3 ($F_{2,12.33} = 3.81$, $P = 0.051$, Fig. 1C) and PC4 ($F_{2,55.00} = 2.49$, $P = 0.09$, Fig. 1D). PC3 loaded positively with red/orange, with *P. nyererei* scoring highest and differing significantly from *P. pundamilia* ($Z = 2.58$, $P = 0.026$), but not quite so from hybrids ($Z = 2.08$, $P = 0.09$). PC4 had a strong, positive association with violet and followed the same general pattern as PC3 (*P. nyererei* highest). There were no significant differences for whole fish PC2 ($P = 0.55$; positive association with green/blue, Fig. 1B). Species differences for each body
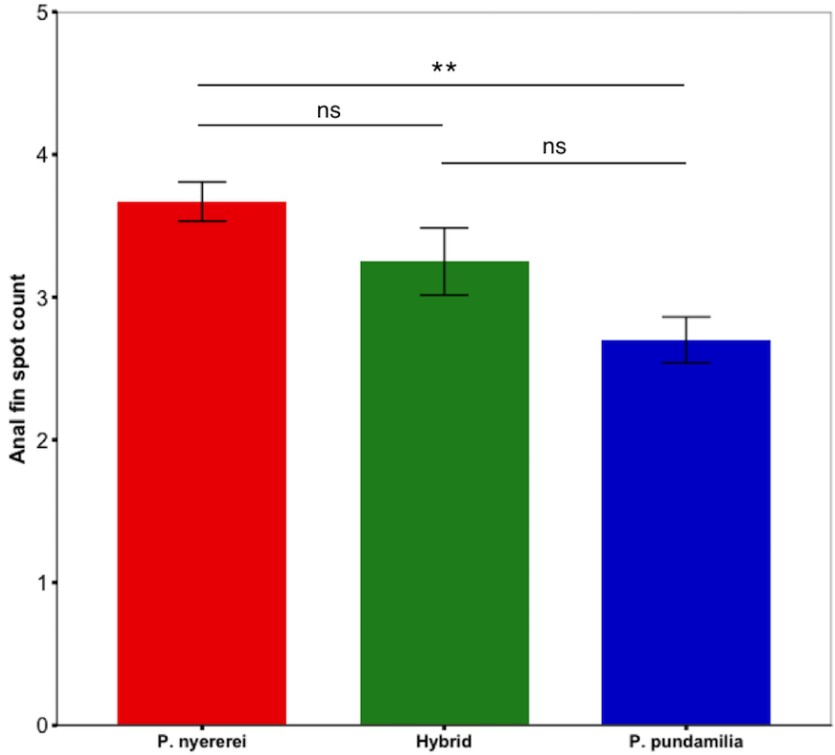

**Figure 2 Species difference in anal fin spot number.** *P. nyererei* had significantly more anal fin spots than *P. pundamilia*, while hybrids were intermediate and did not differ from either parental species. Error bars represent ±one standard error, ** indicates $P < 0.01$.

area separately are presented in Fig. S2. We saw a slight difference in mean brightness ($F_{2,55.00} = 2.5$, $P = 0.08$): *P. nyererei* was lowest, differing somewhat from *P. pundamilia* ($Z = 2.3$, $P = 0.053$), while other comparisons were non-significant ($P > 0.18$).

### Anal fin spots

Anal fin spot coloration did not differ among species (PC1: $P = 0.25$; PC2: $P = 0.15$) but the number of anal fin spots differed significantly ($df = 2$, LRT $= 8.50$, $P = 0.014$; Fig. 2). *P. nyererei* had significantly more spots than *P. pundamilia* ($Z = 2.85$, $P = 0.017$), while hybrids were intermediate and did not differ from either parental species ($P > 0.18$). A statistical trend indicated that anal fin spot brightness also varied between species ($F_{2,55.00} = 2.56$, $P = 0.08$): *P. nyererei* had the brightest spots, differing slightly from hybrids ($Z = 2.17$, $P = 0.07$) but not from *P. Pundamilia* ($P = 0.81$). The total surface area ($P = 0.10$) or the size of the largest anal fin spot did not differ among species ($P = 0.19$).

### Body size

Species differed significantly in SL ($F_{2,55} = 8.06$, $P = 0.008$): hybrids were larger than both *P. nyererei* ($t = 3.50$, $P = 0.002$) and *P. pundamilia* ($t = 3.42$, $P = 0.003$) but the parental species did not differ ($P = 0.98$). There was no relationship between SL and overall fish colorfulness ($P = 0.43$) or anal fin spot coloration ($P > 0.37$). We found
significant, negative relationships between SL and whole fish PC4 ($F_{1,56.00} = 4.95$, $P = 0.03$; strong, positive association with violet), caudal fin PC1 ($F_{1,56.00} = 13.63$, $P < 0.001$; positive with yellow/orange/violet and negative with red/black), and caudal fin PC4 ($F_{1,56.00} = 29.53$, $P < 0.001$; strong, positive loading with violet). Collectively, these results show that smaller fish expressed higher violet colour scores and were generally brighter: brightness was significantly negatively related with SL ($F_{1,56.00} = 11.31$, $P = 0.001$). Violet covered a relatively small proportion of the fish (<l% in *P. pundamilia* & hybrids, ∼2% in *P. nyererei*), while black, whose PC loadings were in the opposite direction of violet (see Table S4), covered a larger area (∼16% in *P. nyererei* & hybrids, ∼7% in *P. pundamilia*). Individual colour analyses revealed a trend for a positive association between SL and black ($F_{1,50.43} = 2.99$, $P = 0.08$), suggesting that larger fish were generally blacker and less bright. Larger fish also had higher total anal fin spot surface area ($F_{1,56.00} = 11.51$, $P = 0.001$).

## Experiment 1: developmental colour plasticity
### No difference in total coloration
Deep- vs. shallow-reared fish did not differ in overall colourfulness or in areas not defined by our colour parameters ($P > 0.5$ for both).

### Increased green in deep light
We predicted that deep-reared fish would increase long-wavelength reflecting coloration, which would imply lower PC1 scores and higher PC3/PC4 scores. However, this was not the case (PC1 & PC3/PC4 scores did not differ between rearing environments, $P > 0.59$ for all). Instead, we found that, independent of species, deep-reared fish had significantly higher PC2 scores ($F_{1,40.07} = 9.08$, $P = 0.004$, Fig. 3A), which could be attributed to body PC2 ($F_{1,40.12} = 4.89$, $P = 0.03$, Fig. 3B) and, to a lesser extent, caudal fin PC2 ($F_{1,30.93} = 3.18$, $P = 0.083$, Fig. 3C). The strongest positive PC2 loadings were with green/blue (body PC2 also loaded positively with red/magenta; caudal fin PC2 with red/violet). We also found a non-significant trend for deep-reared fish to have lower PC4 body scores ($F_{1,56.00} = 3.77$, $P = 0.057$; PC4 loaded negatively for green/black), again indicating increased green colour in deep light. Separate analyses of each colour category confirmed this pattern; only green differed between rearing conditions ($F_{1,40.11} = 11.36$, $P = 0.001$, Fig. 4A). This difference was species-independent, observed in *P. pundamilia*, *P. nyererei*, and hybrids (see Fig. S3). Increased green in deep-reared fish did not correspond to higher brightness ($P = 0.43$). For species-specific coloration in each light environment, see Fig. S4.

### Short vs. long-wavelength colour expression
To test our prediction that deep-reared fish will generally express more long-wavelength colours, we split the measured colours into two categories: reflecting shorter-wavelengths (violet, blue, green) and reflecting longer-wavelengths (yellow, orange, red). This analysis excluded magenta (which has both red and blue components) and black. Contrary to our prediction, deep-reared fish expressed significantly higher amounts of short-wavelength colours ($F_{1,40.12} = 7.40$, $P = 0.009$), while long-wavelength colour expression did not differ ($P = 0.7$).
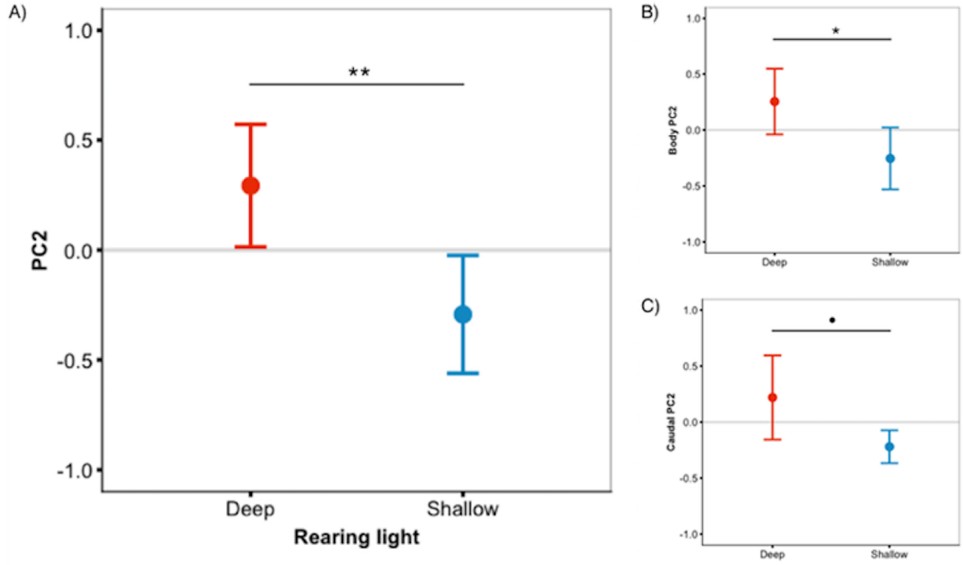

**Figure 3  Deep-reared fish are greener.** (A) Males reared in deep light differed significantly from their shallow-reared brothers in 'whole fish' PC2 scores. These differences could be attributed to the body (B) and, to a lesser extent, the caudal fin (C). Error bars represent 95% CI, ● indicates $P < 0.1$, * indicates $P < 0.05$, ** indicates $P < 0.01$.

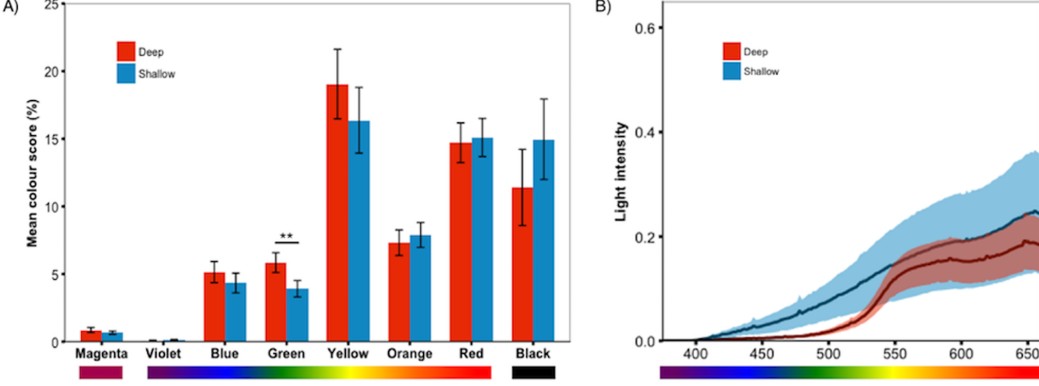

**Figure 4  Treatment effect by colour category.** Analyses of the individual colour scores confirmed the PCA results; (A) deep-reared fish were significantly greener than shallow-reared fish. No other colours differed significantly between rearing environments ($P > 0.23$ for all). Error bars represent ±one standard error, ** indicates $P < 0.01$. (B) The deep and shallow light manipulations differed in the availability of shorter-wavelength light ($\sim$400–550 nm).

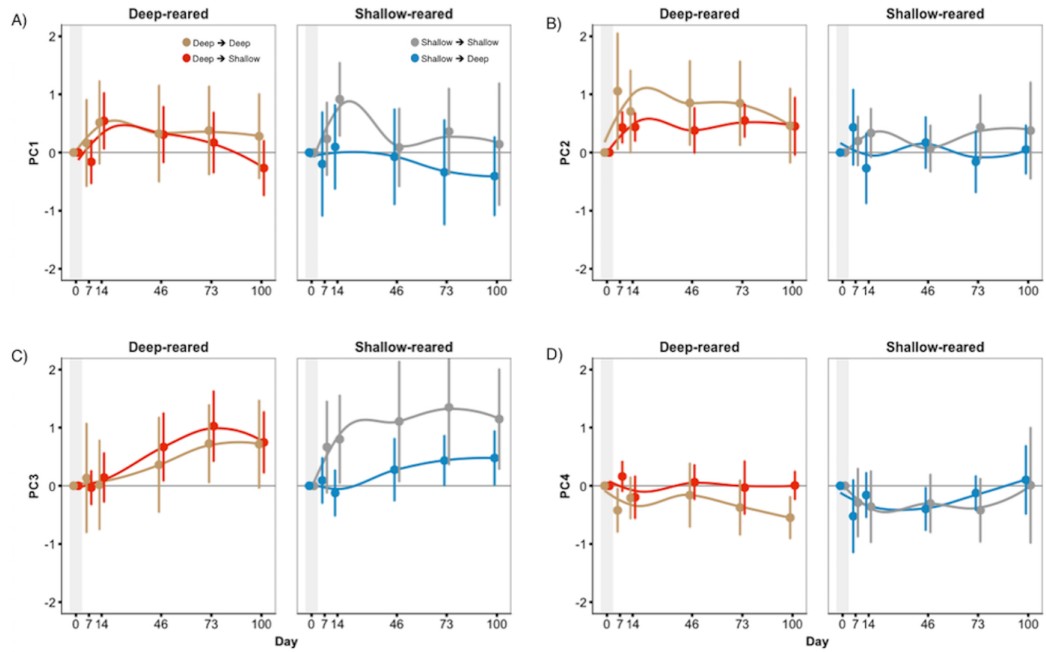

**Figure 5  Little treatment-induced colour change in experiment 2.** Whole fish' PC scores of treatments groups (SD & DS) displayed little difference from control groups (SS & DD) in experiment 2. This was true for (A) PC1, (B) PC2, (C) PC3, and (D) PC4. Scores are presented as the deviation from the mean (zero line) for each fish (three samples each from experiment 1); positive scores indicate an increase in PC scores, while negative indicate a decrease. Error bars represent 95% CI.

### Anal fin spots

Rearing light had no effect on anal fin spot coloration (PC1: $P = 0.30$; PC2: $P = 0.17$), brightness ($P = 0.49$), the number of spots ($P = 0.37$), total surface area ($P = 0.98$), or size of the largest spot ($P = 0.30$).

## Experiment 2: colour plasticity in adulthood
### Little effect of treatment

As seen in Fig. 5, fish that were moved between light conditions did not display consistent changes in coloration compared to baseline or to controls. For whole fish PC3 and for body PC4, we found significant three-way interactions between treatment, species, and date ($F_{6,408} = 2.33$, $P = 0.031$ and $F_{6,408} = 3.34$, $P = 0.003$) but treatment did not cause consistent changes in coloration (Figs. S5 and S6). Treatment had no effect on mean fish brightness ($P = 0.41$) or anal fin spot coloration/brightness ($P > 0.26$). We found a significant effect of 'date' in nearly all analyses (Table S6), indicating that both experimental and control fish continued to change colour over the 100-day sampling period. Thus, the lack of treatment effect was not due to fish colour being inflexible in adulthood.

## DISCUSSION

Local conditions impact the effectiveness of communication signals (*Endler, 1990*; *Endler, 1992*) and can be greatly disrupted by environmental variation. Plasticity in signal

production may be one mechanism to cope with changing conditions. Here, we tested for light-induced plastic changes in the nuptial coloration of *Pundamilia pundamilia and Pundamilia nyererei* by rearing sibling males in environments mimicking deep- and shallow-water habitats of Lake Victoria. We found little evidence for developmental colour plasticity.

## Limited colour plasticity

*P. pundamilia* and *P. nyererei* are naturally depth segregated and occupy different light environments in Lake Victoria (*Maan et al., 2006*; *Seehausen et al., 2008*; *Castillo Cajas et al., 2012*). Given the close proximity of the two habitats (overlapping at some locations), selection might favour some level of flexibility in colour expression to cope with different signaling environments. Previous studies in other fish species have shown light-induced plasticity in coloration (*Fuller & Travis, 2004*; *Lewandowski & Boughman, 2008*; *Hornsby et al., 2013*). Contrary to our predictions, we found that deep-reared fish did not express more long-wavelength reflecting coloration. Instead, deep-reared fish were greener. Our light manipulations did not affect the male colours that most clearly differentiate the two species (blue/red). Additionally, males switched between light environments as adults showed little colour change. We propose two explanations for the lack of plasticity in nuptial coloration.

The first is that nuptial coloration in *Pundamilia* is under strong genetic control. Previous work has shown that species-specific coloration in *Pundamilia* is heritable (*Magalhaes et al., 2009*) and likely controlled by a small number of loci (*Magalhaes & Seehausen, 2010*). Common garden experiments by *Magalhaes et al. (2009)* found higher plasticity in morphological traits than in male colour scores but fish were reared under standard aquarium lighting (for light spectra comparison, see Fig. S1). We observed light-induced plasticity in green, which, unlike other male colours, is not subject to strong divergent selection by female choice (as demonstrated by: *Selz et al., 2014*) and perhaps less rigidly controlled (see below).

A second potential explanation for our results is that our light manipulations mimicked natural spectral variation, but only partially reproduced variation in light intensity. The difference in light intensity between the deep and shallow habitat in Lake Victoria is variable, but the deep habitat is consistently darker. While spectral differences have repeatedly been shown to correlate with numerous *Pundamilia* characteristics, light intensity may also play a role. Future studies could examine this by manipulating light intensity independent of spectral composition.

Finally, our sample size was modest (9–10 and 4 individuals per group in experiment 1 and 2, respectively). However, while increasing the sample size might increase statistical support for some of our results, we found relatively small effect sizes and these would likely not be affected. Thus, we expect that larger sample sizes would not change the main conclusions presented here.

## Increased green in deep

Males reared in the deep light environment were significantly greener than shallow-reared fish. Our light manipulations differed primarily in short-wavelength availability and green
wavelengths were abundant in both conditions (Fig. 4B). If plasticity in the species-specific male colours (blue/red) is limited, then increased green reflectance in darker conditions might be an alternative solution to increase visibility. To test this, we measured mean brightness of fish reared in both conditions. We found no difference in brightness between rearing environments, nor did brightness change when fish were switched in adulthood. This would suggest that differences in green colour do not contribute to increased visibility. However, these results are based on measurements of RGB values from photographs and may not properly capture contrast and perception in a specific light environment. Green covers a relatively small proportion of the fish (∼6% in deep light) and is not concentrated in a specific area of the body (unlike the red dorsum of *P. nyererei,* for example), making reflectance spectrometry difficult. Moreover, changes in green coloration coincided with non-significant changes in multiple other colours (see Fig. 4A), all of which may contribute to detectability.

Contrary to our prediction, we found that deep-reared fish expressed higher total amounts of shorter-wavelength colours, while longer-wavelength colours did not differ. These findings resemble those of *Hornsby et al. (2013)*, who reared Nile tilapia in an environment lacking short-wavelength light and found higher expression of short-wavelength colours. The authors suggested that this response might be adaptive as it increases the contrast against the short-wavelength-poor background (*Lythgoe, 1968*; *Hornsby et al., 2013*). Possibly, this response represents a common strategy in cichlid fish.

### Anal fin spots

Haplochromine cichlids possess carotenoid-dependent, yellow-orange, circular spots on their anal fins (*Goldschmidt, 1991*; *Tobler, 2006*). While the adaptive significance of these spots is debated (*Maan & Sefc, 2013*), previous studies have documented environment-contingent spot coloration in a number of species (*Goldschmidt, 1991*; *Castillo Cajas et al., 2012*; *Theis et al., 2017*). We examined the coloration, brightness, number and size of the anal fin spots and found that none of these measures were influenced by our light manipulations. However, we did find species differences: *P. nyererei* had the highest number of anal fin spots and the spots were generally brighter. Given that *P. nyererei* naturally occurs in the deep, short-wavelength poor habitat, this follows the general patterns presented by *Goldschmidt (1991)* and *Theis et al. (2017)*; the exception being that *P. nyererei* in our study did not exhibit larger fin spots. The absence of colour differences in the anal fin spots of *P. pundamilia* and *P. nyererei* is consistent with earlier results based on reflectance spectrometry of wild fish (*Castillo Cajas et al., 2012*).

### Implications for species isolation

*P. pundamilia* and *P. nyererei* differ in nuptial coloration and colour has been shown to co-vary with light conditions (*Maan, Seehausen & Van Alphen, 2010*; *Castillo Cajas et al., 2012*). Females display divergent preferences for conspecific male colour (*Seehausen & Van Alphen, 1998*; *Haesler & Seehausen, 2005*; *Stelkens et al., 2008*; *Selz et al., 2014*) and these preferences are key to species isolation (*Selz et al., 2014*). Differences in visual system characteristics (*Carleton et al., 2005*; *Maan et al., 2006*; *Seehausen et al., 2008*) correspond

to differences in light environments, male coloration, and female preferences, suggesting a role for divergent sensory drive in speciation (*Maan & Seehausen, 2010*). Recently, we have shown that the same light manipulations that we used here significantly influenced female mate preference, potentially interfering with reproductive isolation (shallow-reared females preferred blue males, while deep-reared females favoured red males (*Wright et al., 2017*)). Plasticity in male colour expression could weaken the linkage disequilibrium between colour and preference even further. However, we find little evidence for such plasticity here, suggesting that blue and red are likely under strong genetic control. This may preserve reproductive isolation between populations inhabiting adjacent visual environments. In contrast, the plastic response in green coloration may aid in overall detectability of males, without interfering with species-assortative mating decisions that rely on interaction at closer range.

## CONCLUSIONS

Our results show that the nuptial coloration of *P. pundamilia* and *P. nyererei* is largely not plastic. Rearing fish in two distinct light conditions mimicking those at different depth ranges in Lake Victoria had little effect on species-specific colour, which is consistent with existing evidence for strong divergent selection on male coloration in this species pair. We did find evidence for light-induced plasticity in green coloration, possibly promoting male detectability but not interfering with species-assortative mating. Taken together, these results provide continued support for the role of the local light environment in species isolation in *Pundamilia*. Reproductive isolation may be affected by environmental change but as this study shows, rapid changes in sexually selected colour signals are unlikely.

## ACKNOWLEDGEMENTS

We acknowledge the Tanzanian Commission for Science and Technology for research permission and the Tanzanian Fisheries Research Institute for hospitality and facilities. We thank Mhoja Kayeba, Mohamed Haluna, Erwin Ripmeester, Oliver Selz, Jacco van Rijssel, Florian Moser, and Joana Meier for help with fish collections and Sjoerd Veenstra and Brendan Verbeek for taking care of the fish in the laboratory. Oliver Selz also provided helpful tips for completing the colour analyses. Demi Damstra and Thomas Scheffers aided in designing and conducting pilot experiments, Titus Hielkema and Silke Scheper helped with image preparation and colour assessment, and Roy Meijer assisted during photography sessions.

### Funding

Financial support came from the Swiss National Science Foundation (SNSF PZ00P3-126340; to Martine E. Maan), the Netherlands Foundation for Scientific Research (NWO VENI 863.09.005; to Martine E. Maan) and the University of Groningen. There was no

additional external funding received for this study. The funders had no role in study design, data collection and analysis, decision to publish, or preparation of the manuscript.

### Grant Disclosures

The following grant information was disclosed by the authors:
Swiss National Science Foundation: PZ00P3-126340.
Netherlands Foundation for Scientific Research: NWO VENI 863.09.005.

### Competing Interests

The authors declare there are no competing interests.

### Author Contributions

- Daniel Shane Wright conceived and designed the experiments, performed the experiments, analyzed the data, contributed reagents/materials/analysis tools, wrote the paper, prepared figures and/or tables, reviewed drafts of the paper.
- Emma Rietveld performed the experiments, analyzed the data, reviewed drafts of the paper.
- Martine E. Maan conceived and designed the experiments, contributed reagents/materials/analysis tools, wrote the paper, reviewed drafts of the paper.

### Animal Ethics

The following information was supplied relating to ethical approvals (i.e., approving body and any reference numbers):

This study was conducted under the approval of the Institutional Animal Care and Use Committee of the University of Groningen (DEC 6205B; CCD 105002016464).

### Field Study Permissions

The following information was supplied relating to field study approvals (i.e., approving body and any reference numbers):

The Tanzania Commission for Science and Technology (COSTECH) approved field permits for the collection of wild fish (2010-100-NA-2010-53 & 2013-253-NA-2014-177).

### Data Availability

Wright DS, Rietveld E, Maan ME. 2017. Replication Data for: Developmental effects of environmental light on male nuptial coloration in Lake Victoria cichlid fish. hdl:10411/Y3GGZS, DataverseNL Dataverse, V1.

### Supplemental Information

Supplemental information for this article can be found online at http://dx.doi.org/10.7717/peerj.4209#supplemental-information.

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
