# Peer review of "Developmental effects of environmental light on male nuptial coloration in Lake Victoria cichlid fish"

_PeerJ, doi:10.7717/peerj.4209_

## Round 0.1 · original submission · Minor Revisions

· Academic Editor

Minor Revisions

We had difficulty getting reviewers on this one, and thus the delay. My apologies. I think you did good work, and reported it well. There are a few minor changes you need to do before we’ll accept the paper. Please follow the reviewer’s and my suggestions; the latter follow immediately.

You measured coloration in P. pundamilia (P), P. nyererei (N), and their hybrids (PN), in two environments that simulated light at two depths of water. This gives six sets of measurements: P, N, and PN for shallow water and P, N, and PN for deep water. I can only guess why you reported only the marginal means (e.g., P, N, & PN averaged over lighting regimes). No matter, please report the results of all six groups before moving on to the marginal means. Can you modify Figs. 1 & 3, so that Fig. 1 included information about the rearing condition (D, S) and Fig. 3 included information about the species (P, N, PN)? Maybe you could add points that show these results?
I had a bit of trouble figuring out how many fish were used in each treatment, and exactly when “things happened”. If the following summary is not correct, you should re-work the Methods section (exps. 1 & 2), and ask a colleague---some biologist who knows nothing about your work, and preferably who works on something unrelated, such as butterfly ecology or bat behavior---to read the revision and tell you what she understands.

Here goes:
You used a total of 58 male fish in the study, for experiment 1. These comprised 6 groups, assigned as follows:
P. pundamilia – shallow 10
P. pundamilia – deep 10
Hybrid – shallow 10
Hybrid – deep 10
P. nyererei – shallow 9
P. nyererei – deep 9
Total 58
You reared these fish until age ~643 days, when you switched 24 of the 58 fish to the opposite treatment, 12 from shallow to deep and 12 from deep to shallow. That left 34 unswitched fish, 18 of which were then placed in different aquaria and 16 of which stayed in their original aquaria.
Now comes a tricky bit. You photographed each of the constant (unswitched) males 3 times, starting at day ~690, at ~2-week intervals. You photographed the switched males 11 times over a 100-day period, but did not photograph these males at the same age as the unswitched fish for one sample point: the second time point for the unswitched group at ~26 days and the eighth time point for the switched group at 46 days. However, the first and third time points for the unswitched group corresponded roughly to the seventh and ninth time points for the switched group
Other minor points:

Please report degrees of freedom as whole numbers. When df’s are estimated because of some deviation from assumptions of a parametric analysis, the resulting numbers are not whole numbers. And, when computer programmers write software for statistical analysis, they may forget to output integers instead of real numbers. Thus, df = 12 rather than df = 12.33 and df = 55 rather than df = 55.00).

In the context of your interspecific crosses, “F1” refers to offspring of P x N or N x P, and not to P x P or N x N, and “F2” refers to offspring of Hybrid x Hybrid (e.g., lines 148 and following)

Table 1 is not necessary.

Reviewer 1 ·

Basic reporting

no comment

Experimental design

no comment

Validity of the findings

no comment

Additional comments

The cichlid species pair Pundamilia pundamilia and P. nyererei represents a well-studied model system to identify mechanisms of ecological speciation. In the present manuscript, Wright et al. investigate environmental light-induced plasticity in nuptial colouration in this cichlid sister species pair from Lake Victoria. To this end, both species (and their hybrids) were reared under light conditions, which mimic their natural habitats, and male nuptial colouration was quantified based on photographs of fish. The study finds very limited plastic responses in male colouration, and importantly not in the species-specific nuptial colours (blue and red), which have been shown to be subject to divergent selection by female choice.
The paper is nicely written, the methods and statistical analyses are sound and the results are generally interpreted adequately. However, a weak point of the study involves the relatively low sample sizes (9 to 10 individuals per species/hybrid and treatment in Experiment 1, and 3 to 4 individuals per species/hybrid and treatment in Experiment 2). This should be addressed in the discussion when interpreting the results.
Please find more specific comments below:

- Presentation of results, lines 288-307. Are the results presented here based on individuals from both shallow and deep ambient light conditions? This is not made clear, and given the aim of the study (lines 130-139), it would make more sense to present the results for each treatment separately (also in Fig. 1 & 2).

- Methods, lines 148-151. Please explain why you used both F1 and F2 crosses.

- Lines 158-160. Differences in body size and species identity between neighbours (and also the number of neighbours) might influence the social rank/dominance among males and thus the expression of nuptial colouration. This was (partly) accounted for in the linear mixed models (line 252), however, whether any of these random effects were significant or not is not mentioned in the results.

- Lines 202-203. Quantifying nuptial colouration in cichlids is notoriously difficult due to the fact that fish change colouration quickly when stressed - which might be the case when transferring the fish into a glass cuvette. How did you standardize photographs, i.e. make sure that you didn’t take pictures of stressed fish with faded colours?

- Please provide legends for supplementary figures and tables.

---

## Round 0.2 · accepted · Accept

· Academic Editor

Accept

Very professional job, guys.

Interesting paper, and remember that my own field of science is far removed from yours.